# Rift Valley fever knowledge, mitigation strategies and communication preferences among male and female livestock farmers in Eastern Province, Rwanda

**Lindsay J. Smith**[1]*, **Janna M. Schurer**[1,2], **Eurade Ntakiyisumba**[3], **Anselme Shyaka**[3ᵒ], **Janetrix Hellen Amuguni**[1ᵒ]

**1** Department of Infectious Disease and Global Health, Cummings School of Veterinary Medicine at Tufts University, North Grafton, Massachusetts, United States of America, **2** Center for One Health, University of Global Health Equity, Kigali, Rwanda, **3** School of Veterinary Medicine, College of Agriculture, Animal Sciences and Veterinary Medicine, University of Rwanda, Nyagatare, Rwanda

ᵒ These authors contributed equally to this work.
* Lindsay.Smith@tufts.edu

**Data Availability Statement:** All relevant data are within the manuscript and its Supporting Information files.

## Abstract

The Government of Rwanda reported an outbreak of Rift Valley fever (RVF) in the Eastern Province in 2018. To respond to the outbreak, vaccination and education campaigns about the disease were carried out. Because RVF cases continue to be detected in Rwanda and the disease impacts livelihoods and health, accurate knowledge and communication are imperative. The objectives of this study were to evaluate knowledge and risk perceptions of RVF transmission among livestock farmers in Nyagatare District, Eastern Province, Rwanda, and to compare RVF knowledge, risk perceptions, and farming practices between male and female livestock farmers. This cross-sectional, quantitative study was conducted in selected sectors of Nyagatare District in the Eastern Province of Rwanda in June 2019. A 34-question survey was used to ask about demographics, livestock ownership, risk perceptions about zoonotic diseases and livestock management, RVF knowledge, preferred communication sources and information sharing strategies, and protective strategies for RVF mitigation while working with livestock. Livestock farmers were interviewed at three milk collection centers, two village meeting points, a farm cooperative meeting, and during door-to-door visits in villages. In total, 123 livestock farmers were interviewed. The survey found that most livestock farmers lacked knowledge about epizootic and zoonotic transmission of RVF, more male livestock farmers were familiar with RVF and risk mitigation strategies, and female livestock farmers are not viewed as reliable sources of information. Additionally, most livestock farmers had not vaccinated their animals against RVF despite past vaccination campaigns. Radio was the most popular communication channel. These findings show that RVF knowledge and information sharing are inadequate among livestock farmers in Eastern Province. Therefore, vaccination and education campaigns may need to be reevaluated within the context of these trends in order to prepare for future RVF outbreaks.

**Funding:** L. S., E. N., and J. H. A. were funded by Cummings Foundation (https://www.cummingsfoundation.org/). J.H. A., J. S., and A. S. were funded by the University of Global Health Equity (https://ughe.org/). A. S. was also funded by the University of Rwanda (UR)-Sweden program (SIDA Research Grant Number 51160027, ORG Prevalence and Risk Factors for the Rift Valley fever in Rwanda) (https://ursweden.ur.ac.rw/). The funders had no role in study design, data collection and analysis, decision to publish, or preparation of the manuscript.

**Competing interests:** The authors have declared that no competing interests exist.

## Author summary

This study was conducted in order to evaluate RVF knowledge and awareness as well as communication and mitigation strategies among livestock farmers in Eastern Province, Rwanda. Rwanda declared an outbreak of RVF in 2018 and cases have continued to be detected. Thus, evaluating the status of knowledge, preventive strategies, and information sharing among livestock farmers is crucial in mitigating future outbreaks. Our team conducted a survey of knowledge, risk perceptions, mitigation strategies, and communication practices among livestock farmers from selected sectors within Nyagatare District and compared them between male and female livestock farmers in order to analyze gender-nuanced differences between these groups. Sectors were chosen for sampling based on their proximity to previous outbreak areas. Survey questionnaire results showed that knowledge and risk perceptions differed between male and female livestock farmers, and that they could be generally improved among all livestock farmers. Female livestock farmers and non-farming community members were viewed as unreliable sources of information which could impact information dissemination. Many farmers also reported that their livestock herds were not vaccinated for the disease despite previous vaccination campaigns. Communication strategies and information sources also differed between male and female livestock farmers, which highlights a need to consider gender in improving RVF vaccination and education campaign coverage. These findings pose implications for future community-based public health interventions as well as policy development for RVF control and mitigating future RVF outbreaks within Rwanda.

## Introduction

Rift Valley fever (RVF) is a mosquito-borne viral zoonotic disease caused by a phlebovirus of the family *Phenuiviridae* [1,2]. Rwanda, a country with extensive cattle production, reported an RVF outbreak in Eastern Province in 2018 [3]. Additional RVF cases detected in Rwanda, with many in Eastern Province, throughout 2020 signify a continuing challenge [4]. While it is difficult to determine exactly when RVF was first detected in Rwanda, according to the OIE World Animal Health Information System, clusters of RVF cases and antibodies had been periodically detected in Rwanda since 2012 [5]. The OIE reported confirmed cases of morbidity and mortality from RVF in livestock in 2012 as well [5]. RVF vaccines were sporadically given in areas of Rwanda between 2012 and 2018; and enzyme-linked immunosorbent assay (ELISA) based detection of antibodies cannot rule out the possibility of antibodies being vaccinal in some instances. Additionally, antibodies can circulate during an interepizootic/epidemic period in animals previously infected with RVF [6,7]. Animals with detected antibodies were likely involved in an enzootic cycle in the interepidemic period, and those with high enough titers were likely protected from the 2018 outbreak [6,7]. Although serology has detected RVF antibodies and RVF morbidity and mortality have been confirmed in Rwanda since 2012 and reported by the OIE, an RVF outbreak was not officially reported by the Government of Rwanda until 2018 [5]. In addition to sporadic vaccination, vaccination campaigns also occur annually in areas where RVF outbreaks have been reported in order to control the disease and mitigate impacts on the livestock sector. The vaccination is voluntary and livestock farmers may be requested to pay a fee to get their cattle vaccinated. Currently, Rwanda uses a live attenuated RVF vaccine manufactured by Biopharma in Morocco. In addition to vaccination, other factors including decreased vectors in the dry season and gradually decreasing RVF circulation

levels during interepidemic periods likely contribute to decreased case numbers between outbreaks [8].

In 2018, livestock RVF cases were reported in eight sectors across five districts (Ngoma, Kirehe, Rwamagana, Kayonza, Gatsibo) [9,10,11]. Hundreds of cattle aborted and cases were confirmed via ELISA in the national veterinary laboratory of Rubirizi [3,9,10]. The Rwanda Agriculture Board responded by vaccinating 237,386 cattle, 22,727 goats, and 17,872 sheep against RVF [5]. Animal movements were banned from mid-June to the end of July to mitigate the outbreak. The government implemented vector control, ante- and post-mortem inspections, and public awareness campaigns [9,10].

Throughout 2020, Rwanda experienced 32 outbreaks of RVF across the country with 689 total cases recorded in ruminant livestock [4]. Nyagatare experienced three outbreaks in January and one in March, with 213 total cases [4]. In other parts of Eastern Province, specifically Gatsibo and Ngoma Districts, a total of 109 RVF cases were recorded in ruminant livestock species [4]. In response to these RVF outbreaks in 2020, a total of 354,380 animals were vaccinated for RVF around outbreak areas [4].

Cattle have great cultural and economic significance in Rwanda while sheep and goats are widespread and have potential to advance livestock sector development. Traditionally, marriage in Rwanda is marked by the presentation of a bride dowry in the form of cattle. The Girinka Program, or the One Cow per Family Program, provides a free cow to a poor family to reduce childhood malnutrition and raise rural incomes through milk sales [12]. Additionally, 11% of Rwanda's gross domestic product (GDP) came from the livestock subsector in 2019 [13–15]. Small ruminants have been reported to suffer higher mortality from RVF than cattle, and sheep and goat meat production are projected to increase by 33% and 50% by 2021/22, respectively [16,17].

Livestock farmer behaviors and risk perceptions are crucial in controlling zoonotic diseases. During RVF outbreaks in South Africa, Tanzania, and Kenya, most human and animal RVF cases were attributed to livestock farmers handling contaminated fomites and infective tissues or consuming infected livestock products rather than to mosquito bites [1,2,18,19]. Thus, in order to mitigate future RVF outbreaks in Rwanda, the behavioral risk perceptions of livestock farmers must be understood. Studies show that livestock farmers and butchers in East African countries are likely to recognize that they are at risk for contracting RVF, but many do not know the RVF symptoms seen in humans or animals nor wear personal protective equipment [20–23]. Additionally, male and female livestock farmers have different farming experiences, and understanding gender differences in knowledge and risk perceptions can maximize intervention efficacy.

As a result of climate change and shifting mosquito ranges, RVF is spreading to new environments while threatening rural livelihoods and health [24]. Understanding behavioral risk perceptions of RVF among high risk populations like livestock farmers can provide local and national governments with information about who and what to target within effective educational campaigns. There are no published studies of livestock farmer knowledge or risk perceptions regarding RVF in Rwanda. Thus, the specific aims of this study were to (1) evaluate knowledge and risk perceptions of RVF among livestock farmers in Nyagatare District, Eastern Province, Rwanda, and (2) to compare RVF knowledge, risk perceptions, and farming practices between male and female livestock farmers there.

## Methods

### Ethics statement

Verbal consent was obtained from participants prior to every interview. The data were analyzed anonymously. Ethical approval for this study was granted by the Institutional Review

Board at Tufts University, Massachusetts, United States of America, (#1955018) and by the Research Screening and Ethical Clearance Committee of the College of Agriculture, Animal Sciences, and Veterinary Medicine at the University of Rwanda (Ethical approval reference: 025/17/DRIPGS).

### Description of the study area

This study was conducted in four sectors of Nyagatare District in the Eastern Province of Rwanda, with most participants living in Rwimiyaga and Rwempasha, and others living in Nyagatare and Rukomo sectors (Fig 1). These sectors were chosen based on their proximity to sectors with previously recorded RVF outbreaks, level of cattle production, and proximity to rice wetlands. Rwimiyaga reported cases of RVF in 2018, while Rwempasha did not. Nyagatare and Rukomo were unaffected but are located close to outbreak areas; and some livestock farmers at milk collection centers in Rwimiyaga and Rwempasha met the inclusion criteria and reported living in Nyagatare and Rukomo sectors [5,9]. The estimated total human population of Nyagatare District, using data from the 2011 Integrated Household Living Conditions Survey (EICV3) and national population medium-level growth projections between 2012 and

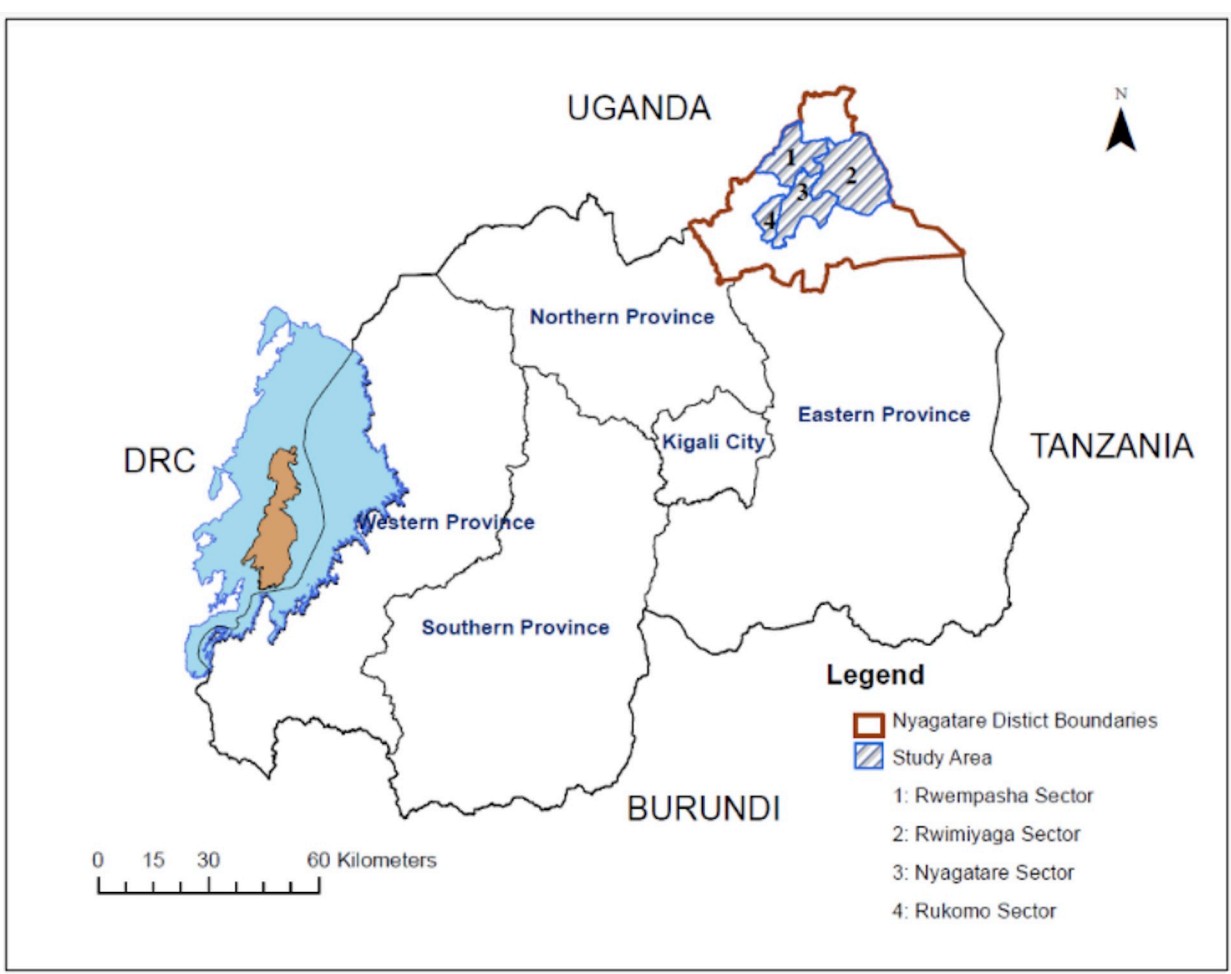

**Fig 1. Map of study area.** This map was created using QGIS Ver. 10.5. The layers are freely accessible from https://www.diva-gis.org/datadown and they can be shared under CC-BY license 4.0.

2020, is approximated to be 512,200 [25]. Nyagatare District is comprised of 14 sectors in total. Using sector population estimates from 2012 and the medium-level projected national population growth for Rwanda between 2012 and 2020, it can be estimated that the population of Rwimiyaga is now approximately 69,000, Rwempasha is 24,600, Nyagatare is 62,500, and Rukomo is 41,000 [26]. The largest source of employment in Nyagatare is agriculture with 79.6% of men and women aged 16 years or older engaged in cash crop or livestock production (primarily cattle, goats, sheep, and chickens) [26]. Most men and women in Nyagatare are small-scale farm owners (57% and 74.5%, respectively) or hourly waged farm workers (13% each) [16]. Cattle production systems are extensive or semi-intensive with low levels of intensification [16]. Nyagatare District and its corresponding cattle grazing areas are located adjacent to extensive stagnant water in rice wetlands, which can serve as mosquito breeding sites [1,2,5]

## Study design and data collection

A cross-sectional, quantitative study was conducted in June 2019 and data were collected via a quantitative survey that was conducted via interview. Using a convenience sampling strategy, the research team sampled exhaustively at every sampling location by approaching prospective participants at three milk collection centers in Rwimiyaga, Rwempasha and Nyagatare, two village meeting points in Rwempasha, a farm cooperative meeting in Rwimiyaga, and during door-to-door visits in villages in Rwempasha until no additional participants could be found within each sampling time period (Table 1). The sample size was a convenience sample. The size and scope of our convenience sample was influenced by time and budget. A true estimate of livestock farmers within and their distribution across the four sectors is not available and the livestock farmers we interviewed were only a subset of livestock farmers living in Eastern Province. In total, 123 livestock farmers from four sectors of Nyagatare District were interviewed. The inclusion criteria were ruminant livestock ownership and being an adult that was 18 years of age or older. Any livestock farmer meeting the inclusion criteria was interviewed. In terms of overall relative gender distribution, more male livestock farmers were interviewed at milk collection centers and more female livestock farmers were interviewed in door-to-door visits, village meeting points, and cooperative meetings. If a family was encountered that had both female and male members present, the person primarily responsible for livestock caretaking was interviewed.

The questionnaire included 34 questions and six parts: a) demographics, b) livestock ownership, c) risk perceptions about zoonotic diseases and livestock, d) RVF knowledge e) preferred communication sources and information sharing strategies, and f) protective strategies for RVF mitigation while working with livestock. It was written in English and then translated to Kinyarwanda, with interviews conducted in Kinyarwanda. A mixture of yes/no, open-ended, and option selection questions were used. Two "yes" or "no" questions focusing on specific RVF transmission knowledge were only asked to respondents who had already heard of RVF. Follow-up questions were asked to participants to verify their knowledge of transmission routes.

**Table 1. Number of participants interviewed at each meeting location.**

| Meeting location | Administrative locality (area) | Number of participants |
|---|---|---|
| Milk collection center | Rwimiyaga, Rwempasha, Nyagatare | 62 |
| Village meeting point | Rwempasha | 19 |
| Cooperative meeting | Rwimiyaga | 22 |
| Door-to-door | Rwempasha | 20 |

The questionnaire was pre-tested on a group of six volunteers from Nyagatare town and adjusted as needed before it was uploaded onto mobile phones using the Kobo Toolbox application (https://www.kobotoolbox.org/). Livestock farmer responses were directly entered into KoboToolbox in English, uploaded to the server, and checked for errors daily. Data collection was completed by two veterinarians and a veterinary student. Each were trained to collect interview data and enter it into KoboToolbox.

## Statistical analysis

Frequencies for responses were calculated using SPSSv26 (IBM, United States) statistical software. Reponses to all questions were reported descriptively. Fisher's Exact Test was used to test for association between respondent gender or administrative locality and respondent knowledge or communication preferences. The cutoff for statistical significance was 5%. Ages were divided into quintiles.

## Results

### Participant demographics

In this study, 62 (50.4%) women and 61 (49.6%) men were interviewed, with most participants aged between 22 and 53 years (35.8% between 22 and 37, 38.2% between 38 and 53; Table 2). Participants most often owned cattle and/or goats (92.7%) versus other animals such as chickens, pigs, sheep, dogs, cats, and rabbits. Three-quarters of participants (75.6%) had no education or had completed primary school; two-thirds (66.7%) had raised livestock for over 10 years.

### Knowledge and risk perceptions of Rift Valley fever

Overall, more male participants (78.7%) had heard of RVF than female participants (59.7%, p = 0.031) and more men than women had received advice on minimizing the risk of infection (37.7% versus 12.9%; p = 0.002; Table 3). Less than half were aware that people could be infected (43.0%) and few understood how livestock were infected (15.4%). Upon answering follow-up questions, approximately half of participants (43.9%) could name specific zoonotic diseases (Table 3). Most frequently named zoonotic diseases were foot and mouth disease, tuberculosis, brucellosis, anthrax, and intestinal parasites, and of the participants who named zoonotic diseases, only 5% named RVF. Educational level was statistically associated with perception of risk associated with livestock rearing (p = 0.020).

### Risk mitigation efforts for Rift Valley fever

Approximately half of participants (44.7%) believed that tending livestock posed health risks and 25.2% took measures to protect themselves while working with animals (Table 4). More men slaughtered their livestock than women (32.8% versus 12.9%, p = 0.010) and the majority (92.9%) of respondents who slaughtered their own animals reported not wearing personal protective equipment when slaughtering animals. Only one in four participants (22%) had vaccinated livestock against RVF (Table 4). Educational level was not associated with using protective measures (p = 0.267)

### Communication practices and preferences

Most participants (88.6%) heard about disease outbreaks on the radio (Table 5). Respondents most often preferred obtaining information on the radio (33%), followed by the telephone (27.3%), face-to-face conversations (23.4%), newspapers (1.9%) and other (14.4%), which

**Table 2. Demographic profile of respondents from Nyagatare District, Rwanda (N = 123).**

| Variable | Frequency n (%) |
|---|---|
| **Gender** | |
| Female | 62 (50.4) |
| Male | 61 (49.6) |
| **Age quintiles (years)** | |
| 22–37 | 44 (35.8) |
| 38–53 | 47 (38.2) |
| 54–69 | 28 (22.8) |
| 70–85 | 2 (1.6) |
| >85 | 2 (1.6) |
| **Education Level** | |
| No formal education | 38 (30.9) |
| Primary | 55 (44.7) |
| Secondary | 26 (21.1) |
| University | 4 (3.3) |
| **Sector**[1] | |
| Nyagatare | 13 (10.6) |
| Rukomo | 1 (0.8) |
| Rwempasha | 45 (36.6) |
| Rwimiyaga | 64 (52.0) |
| **Years Raising Livestock** | |
| <1 | 8 (6.5) |
| 1–5 | 24 (19.5) |
| 6–10 | 9 (7.3) |
| > 10 | 82 (66.7) |
| **Animals Owned**[2] | |
| Cattle | 114 (92.7) |
| Goat | 64 (52.0) |
| Chicken | 22 (17.9) |
| Sheep | 14 (11.4) |
| Pig | 3 (2.4) |
| Other[3] | 3 (2.4) |

[1] While Rwimiyaga and Rwempasha were the two main target sectors for this study, some livestock farmers at milk collection centers who met the inclusion criteria reported being from Nyagatare and Rukomo sectors.

[2] Respondents could own multiple animal types

[3] Other- dogs, cats, rabbits

included community meetings, veterinarians, cooperatives, and community health workers. There were no significant differences between men and women with respect to communication preferences.

## Attitudes toward source of information

More women (83.9%) than men (63.9%) felt women were credible sources of advice about crop production and raising livestock (p = 0.014). Approximately half (52%) of respondents felt that non-livestock farmers could provide credible crop production- and livestock-related information (Table 6). There was no significant difference between the attitudes of male and female livestock farmers toward the credibility of information from non-livestock farmers.

**Table 3. Knowledge and risk perceptions of Rift Valley fever among participants from Nyagatare District, Rwanda (N = 123).**

| | Male | Female | Total | Fisher Test p-value |
|---|---|---|---|---|
| | | n (%) | | |
| Has heard of RVF | | | | |
| Yes | 48 (78.7) | 37 (59.7) | 85 (69.1) | |
| No | 13 (21.3) | 25 (40.3) | 38 (30.9) | 0.031 |
| Knows humans can be infected with RVF (n = 85) | | | | |
| Yes | 20 (41.7) | 17 (45.9) | 37 (43.5) | |
| No | 28 (58.3) | 20 (54.1) | 48 (56.5) | 0.778 |
| Knows how livestock become infected with RVF (n = 85) | | | | |
| Yes | 11 (22.9) | 8 (21.6) | 19 (22.4) | |
| No | 37 (77.1) | 29 (78.4) | 66 (77.6) | 0.927 |
| Has received advice on minimizing risk of RVF | | | | |
| Yes | 23 (37.7) | 8 (12.9) | 31 (25.2) | |
| No | 38 (62.3) | 54 (87.1) | 92 (74.8) | 0.002 |
| Believes tending livestock poses health risks | | | | |
| Yes | 30 (49.2) | 25 (40.3) | 55 (44.7) | 0.367 |
| No | 31 (50.8) | 37 (59.7) | 68 (55.3) | |
| Is aware of specific zoonotic diseases transmissible from livestock to people | | | | |
| Yes | 35 (57.4) | 32 (51.6) | 67 (54.5) | |
| No | 26 (42.6) | 30 (48.4) | 56 (45.5) | 0.588 |

## Rift Valley fever knowledge across Rwimiyaga and Rwempasha sectors

Participants differed between sectors in zoonosis awareness (p = 0.007), with 36.7% of respondents from Rwimiyaga saying livestock can transmit disease, versus 14.7% of respondents from Rwempasha. When faced with ill livestock, Rwimiyaga participants were more likely to call a veterinarian than Rwempasha participants (43.1% and 23.9% respectively, p = 0.030). Respondents who were not likely to call a veterinarian when their animals got sick reported purchasing tetracycline at a pharmacy and treating their animals with this antibiotic themselves. There was no significant difference between sectors regarding whether participants had

**Table 4. Risk mitigation efforts for Rift Valley fever among participants from Nyagatare District, Rwanda (N = 123).**

| | Male | Female | Total | Fisher Test p-value |
|---|---|---|---|---|
| | | n (%) | | |
| Wears personal protective equipment when slaughtering animals | | | | |
| Yes | 2 (10.0) | 0 | 2 (7.1) | |
| No | 18 (90.0) | 8 (100.0) | 26 (92.9) | 1.00 |
| Slaughters his/her own animals | | | | |
| Yes | 20 (32.8) | 8 (12.9) | 28 (22.8) | |
| No | 41 (67.2) | 54 (87.1) | 95 (77.2) | 0.010 |
| Takes protective measures while working with animals | | | | |
| Yes | 18 (29.5) | 13 (21.0) | 31 (25.2) | |
| No | 43 (70.5) | 49 (79.0) | 92 (74.8) | 0.305 |
| Animals are vaccinated against RVF | | | | |
| Yes | 18 (29.5) | 9 (14.5) | 27 (22.0) | 0.052 |
| No | 43 (70.5) | 53 (85.5) | 96 (78.0) | |

**Table 5. Preferred sources of information among respondents in Nyagatare District, Rwanda (N = 123).**

| | Male | Female | Total | Fisher Test p-value |
|---|---|---|---|---|
| | n (%) | | | |
| Hears about disease outbreaks on the radio | | | | |
| Yes | 55 (90.1) | 54 (87.1) | 109 (88.6) | |
| No | 6 (9.9) | 8 (12.9) | 14 (11.4) | 0.778 |
| What is your preferred method(s) of receiving information?[3] | | | | |
| Radio | 32 (29.4) | 37 (37.0) | 69 (33.0) | |
| Telephone | 30 (27.5) | 27 (27.0) | 57 (27.3) | |
| Talking in person | 30 (27.5) | 19 (19.0) | 49 (23.4) | |
| | | | | 0.129 |
| Newspapers | 1 (0.9) | 3 (3.0) | 4 (1.9) | |
| Other[4] | 16 (14.7) | 14 (14.0) | 30 (14.4) | |

[3]Respondents could select more than one preferred method of receiving information.

[4]Other forms of communication included community and farmer meetings, television, livestock farmers, veterinarians, community health workers, someone with training on farming techniques, and cooperatives.

heard of RVF, knew if people can be infected with RVF, or knew how livestock become infected with the disease. Sector was also not significantly related to wearing personal protective equipment when slaughtering animals or having ever had animals vaccinated for RVF.

## Discussion

RVF outbreaks continue to occur in Rwanda while threatening livestock health and production, human health, and rural livelihoods. Gendered dimensions of information uptake as well as educational and vaccination campaigns are important to consider in designing interventions to control the disease. While RVF knowledge has been evaluated in other countries in Africa that have experienced larger and more frequent outbreaks, it has not been evaluated in Rwanda which has continued to detect cases and experience outbreaks throughout the last nine years. Establishing a baseline of RVF awareness and knowledge among livestock farmers in Rwanda helps inform interventions. The low levels of RVF knowledge and vaccination coverage reported among livestock farmers as well as differences in knowledge and information sharing among male and female livestock farmers in our study highlight necessary considerations in mitigating future impacts of the disease in Rwanda.

Because many livestock farmers reported being familiar with RVF but did not have accurate knowledge about epizootic or zoonotic transmission, they may be unaware of their lack of knowledge, and because of this they may not seek nor be receptive to more accurate, up-to-

**Table 6. Attitudes toward source of information among respondents in Nyagatare District, Rwanda (N = 123).**

| | Male | Female | Total | Fisher Test p-value |
|---|---|---|---|---|
| | n (%) | | | |
| Thinks women are credible sources of advice | | | | |
| Yes | 39 (63.9) | 52 (83.9) | 91 (74.0) | |
| No | 22 (36.1) | 10 (16.1) | 32 (26.0) | 0.014 |
| Thinks non-livestock farmers can be credible sources of information | | | | |
| Yes | 29 (47.5) | 35 (56.5) | 64 (52.0) | |
| No | 32 (52.5) | 27 (43.5) | 59 (48.0) | 0.369 |

date information [27,28]. Additionally, low levels of awareness and knowledge have been identified as results of communication inequality and accessibility, which alter exposure to and success of public health messaging [27,29]. Thus, accessibility, frequency, and distribution of education campaigns could create barriers to information dissemination in communities, especially in rural areas and villages in Eastern Province. Experience with active RVF outbreaks could also influence retention of accurate, thorough RVF knowledge, and the relatively smaller number of outbreaks in Rwanda compared to other countries could reduce overall knowledge of the disease. A study on knowledge, attitudes, and practices among farmers and slaughterhouse workers in Uganda found that while participants recognized RVF, less than half could accurately describe transmission between animals and from animals to humans [29]. However in Kenya, where larger and more frequent RVF outbreaks have occurred with significant economic impacts, study participants have demonstrated more thorough knowledge of RVF [30,31].

Despite past vaccination campaigns in Rwanda, many livestock farmers surveyed in this study reported that their animals were not vaccinated for RVF. Vaccination campaigns may not be sufficiently covering the Eastern Province, which could be due to financing, supply chain, and policy challenges [32]. Inadequate infrastructure for distribution, cold chain, and transport could lead to decreased vaccine penetration [32]. Stockpiling of vaccines and limited supply can affect coverage as well, and logistics of vaccination campaign rollout, such as locations targeted in vaccination campaigns as well as timing and methods of information sharing, can limit the number of farmers who take their livestock to get vaccinated [32,33]. Continued RVF outbreaks could place more strain on the supply of vaccines and lead to limited coverage during vaccination campaigns.

Differences between men and women in RVF knowledge, understanding of RVF zoonotic transmission, and receiving advice on RVF risk mitigation show different levels of access to information. In Rwanda, men manage cows and cash crops while women manage small animals such as goats and chickens [34]. Additionally, RVF education campaigns in Rwanda are done prior to when livestock, mostly cattle, get vaccinated. Cattle keepers and owners, the vast majority of whom are men, bring cows for vaccination and thus they are more likely to benefit from the RVF education campaigns and know more about the disease. Furthermore, 79.1% of women are involved in agriculture, with most being subsistence agriculture, and few women belong to agricultural cooperatives [35]. Women's household labor responsibilities, their limited livestock ownership rights, decision-making power, and control over income make them less likely to work with cattle and therefore less likely to gain knowledge and vaccinations from RVF campaigns [36–39]. Additionally, more male farmers reported slaughtering their own animals, thus they may be more knowledgeable of RVF since they are more directly at risk with this activity [28,30]. The structure of intra-household decision-making and income control presents barriers to vaccination and information uptake at the community level, which has implications for long-term RVF control in Rwanda [37–42]. In past educational campaigns, RVF information was informally presented before, while, and after administering vaccines, with local veterinarians reporting numbers of cattle vaccinated and money paid to vaccination assistants. No further documentation is produced. If women miss educational campaigns, participants and species vaccinated are not documented, and women do not discuss RVF with male farmers, knowledge and vaccination gaps will persist.

Knowledge about RVF and its transmission needs to improve in Eastern Province so that livestock farmers can protect themselves and their animals from outbreaks by pursuing vaccination. Additionally, improving information dissemination about vaccination campaigns can increase vaccination penetration. Gendered approaches to radio-based information dissemination should be considered, as radio has been found to be preferred and effective. Gendered

differences in livestock ownership and information access should also be evaluated to improve RVF knowledge and vaccination. For example, female livestock farmers tend to own small ruminants like goats and sheep instead of cattle [34,36–39]. With cattle and male cattle owners and keepers being primarily targeted during campaigns, small ruminants will remain susceptible to RVF and also spread the virus to cattle. Female livestock farmers' lack of knowledge can expose them to RVF as a zoonosis and enhance transmission of RVF between small ruminants and cattle, especially as small ruminants are less likely to be vaccinated. Livestock farmers with larger herds may be more likely to pursue RVF education and vaccination, and smaller herds as well as goats and sheep may be missed in campaigns and continue to circulate the virus [17,20,21]. Finally, because radio is so popular and accessible to both male and female livestock farmers, information shared on other communication channels may not be as successful in reaching livestock farmers, or the current frequency of information dissemination via radio may need to increase in order to improve knowledge retention among livestock farmers [43].

There were several limitations to this study. First, our study population was relatively small. Analyses were done with a Fisher's Exact Test due to the small sample size since it is the more appropriate statistical test for analyzing data from small sample sizes [44,45]. This study was a pilot study that should be scaled up to a larger sample size and replicated in the future. Additionally, this study compared respondents in a small geographic area in Rwanda. Furthermore, this study was designed to only include a survey questionnaire to obtain an assessment of the status of RVF knowledge and awareness specifically among livestock farmers in Eastern Province, Rwanda. It was not designed to conduct open-ended, in-depth interviews or focus groups or key informant interviews with ministry officials. While some survey questions were open-ended, it could be beneficial for additional studies in Rwanda to implement in-depth interviews and focus groups to help further explain the reasons for the status of RVF knowledge and herd vaccination in Eastern Province. Our team asked about information sharing but did not ask specifically about how information was spread from livestock farmers and keepers who participated in vaccination and education campaigns to other family and community members. Additional studies could determine how information spreads throughout families and communities following campaigns. Lastly, because this was a survey-based study asking respondents about their knowledge and practices to prevent RVF transmission, there could be bias in answers due to respondents wanting to appear responsible.

## Conclusions and recommendations

Our study of livestock farmers living adjacent to RVF outbreak areas highlighted generally poor knowledge of RVF, poor vaccination coverage, significant differences between men and women in RVF knowledge and risk mitigation, different preferred sources of information among men and women, and radio being the most preferred communication channel. The differences between male and female livestock farmers highlight gendered dimensions of knowledge uptake and information sharing, and are important considerations for future interventions.

RVF educational campaigns should be evaluated on the ground to ensure that information is disseminated to both women and men. Further documentation of campaign participant demographics and information disseminated should be required to ensure adequate coverage. Targeted interventions maximize potential for behavior change, which is instrumental in mitigating RVF outbreaks.

Educational campaigns can encourage vaccination, vector control, and personal protection, and serve as the first step in mitigating zoonotic disease impacts within Eastern Province. On a global scale, this will advance the Sustainable Development Goals of reducing poverty, eradicating hunger, promoting good health, and improving gender equality.

## Supporting information

**S1 RVF Survey Questionnaire. This questionnaire includes all of the survey questions that livestock farmers were asked during interviews in the field in order to collect data for this project.**
(DOC)

## Acknowledgments

The authors wish to acknowledge Nyagatare district and sector veterinary officers, community leaders and members, community health workers, and Radio Nyagatare. They sincerely thank Hyacinthe Dusingize for her significant contributions to survey questionnaire development and data collection, as well as for conducting surveys. Additionally, they extend gratitude to the Cummings School of Veterinary Medicine, the University of Rwanda College of Agriculture, Animal Sciences, and Veterinary Medicine, and the Tufts-UGHE-UR One Health Collaborative for supporting this project and aiding in its publication.

## Author Contributions

**Conceptualization:** Lindsay J. Smith, Janna M. Schurer, Eurade Ntakiyisumba, Anselme Shyaka, Janetrix Hellen Amuguni.

**Data curation:** Lindsay J. Smith.

**Formal analysis:** Lindsay J. Smith.

**Funding acquisition:** Anselme Shyaka, Janetrix Hellen Amuguni.

**Investigation:** Lindsay J. Smith, Janna M. Schurer, Eurade Ntakiyisumba, Anselme Shyaka.

**Methodology:** Lindsay J. Smith, Janna M. Schurer, Eurade Ntakiyisumba, Anselme Shyaka.

**Project administration:** Lindsay J. Smith, Eurade Ntakiyisumba, Anselme Shyaka, Janetrix Hellen Amuguni.

**Resources:** Anselme Shyaka, Janetrix Hellen Amuguni.

**Software:** Lindsay J. Smith, Eurade Ntakiyisumba, Anselme Shyaka.

**Supervision:** Anselme Shyaka, Janetrix Hellen Amuguni.

**Validation:** Lindsay J. Smith, Anselme Shyaka.

**Visualization:** Lindsay J. Smith, Eurade Ntakiyisumba.

**Writing – original draft:** Lindsay J. Smith.

**Writing – review & editing:** Lindsay J. Smith, Janna M. Schurer, Eurade Ntakiyisumba, Anselme Shyaka, Janetrix Hellen Amuguni.

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
